# Validation of a spatial agent-based model for *Taenia solium* transmission ("CystiAgent") against a large prospective trial of control strategies in northern Peru

Ian W. Pray[1]*, Francesco Pizzitutti[1], Gabrielle Bonnet[1], Eloy Gonzales-Gustavson[2,3], Wayne Wakeland[4], William K. Pan[5], William E. Lambert[1], Armando E. Gonzalez[3], Hector H. Garcia[6,7], Seth E. O'Neal[1,7], for the Cysticercosis Working Group in Peru.

1 School of Public Health, Oregon Health & Science University and Portland State University, Portland, Oregon, United States of America, 2 Tropical and Highlands Veterinary Research Institute, School of Veterinary Medicine, Universidad Nacional Mayor de San Marcos, EL Mantaro, Peru, 3 School of Veterinary Medicine, Universidad Nacional Mayor de San Marcos, Lima, Peru, 4 Systems Science Program, Portland State University, Portland, Oregon, United States of America, 5 Duke Global Health Institute & Nicholas School of Environment, Duke University, Durham, North Carolina, United States of America, 6 School of Sciences, Department of Microbiology, Universidad Peruana Cayetano Heredia, Lima, Peru, 7 Center for Global Health Tumbes, Universidad Peruana Cayetano Heredia, Tumbes, Peru

* ian.pray@gmail.com

## Abstract

### Background

The pork tapeworm (*Taenia solium)* is a parasitic helminth that imposes a major health and economic burden on poor rural populations around the world. As recognized by the World Health Organization, a key barrier for achieving control of *T. solium* is the lack of an accurate and validated simulation model with which to study transmission and evaluate available control and elimination strategies. CystiAgent is a spatially-explicit agent based model for *T. solium* that is unique among *T. solium* models in its ability to represent key spatial and environmental features of transmission and simulate spatially targeted interventions, such as ring strategy.

### Methods/Principal findings

We validated CystiAgent against results from the Ring Strategy Trial (RST)–a large cluster-randomized trial conducted in northern Peru that evaluated six unique interventions for *T. solium* control in 23 villages. For the validation, each intervention strategy was replicated in CystiAgent, and the simulated prevalences of human taeniasis, porcine cysticercosis, and porcine seroincidence were compared against prevalence estimates from the trial. Results showed that CystiAgent produced declines in transmission in response to each of the six intervention strategies, but overestimated the effect of interventions in the majority of villages; simulated prevalences for human taenasis and porcine cysticercosis at the end of the

**Data Availability Statement:** All relevant data are within the manuscript and its Supporting Information files.

**Funding:** This research was funded in part by the US National Institute of Allergy and Infectious Disease (R01AI141554), the National Institute of Neurologic Disorders and Stroke (R01NS080645), and the Fogarty International Center through the National Institutes of Health. Grants were awarded to SEO as principal investigator. SEO, FP, GB, EGG, WP, AEG and HHG receive salary through these grants. IWP was supported with travel and stipend for doctoral work by the Fulbright US Students Program and Fulbright Commission in Peru. The funders had no role in study design, data collection and analysis, decision to publish, or preparation of the manuscript.

**Competing interests:** The authors have declared that no competing interests exist.

trial were a median of 0.53 and 5.0 percentages points less than prevalence observed at the end of the trial, respectively.

## Conclusions/Significance

The validation of CystiAgent represented an important step towards developing an accurate and reliable *T. solium* transmission model that can be deployed to fill critical gaps in our understanding of *T. solium* transmission and control. To improve model accuracy, future versions would benefit from improved data on pig immunity and resistance, field effectiveness of anti-helminthic treatment, and factors driving spatial clustering of *T. solium* infections including dispersion and contact with *T. solium* eggs in the environment.

## Author summary

Neurocysticercosis, caused by the ingestion of *Taenia solium* eggs, is a major cause of human epilepsy around the world. A wide spectrum of tools to fight *T. solium* is are now available and include antiparasitic treatment for pigs and humans, porcine vaccines, and sanitation improvements; however, the ideal combination of interventions applied to populations to maximize effectiveness and feasibility is not known. Transmission models are one tool that can be used to compare and evaluate different intervention strategies, but no currently available *T. solium* models have been tested for accuracy. In this research, we validated our model ("CystiAgent") by comparing simulations of the model to the results of a large-scale trial testing a variety of *T. solium* control interventions. The model was calibrated using observed epidemiological data from these villages and evaluated for its ability to reproduce the effect of *T. solium* control interventions. The validation showed that the model was able to reproduce the baseline levels of disease, but generally overestimated the effect that each intervention would have on transmission. These results will allow us to identify limitations of the current model to improve future versions, and represent a step forward in the creation of a tool to design and evaluate future programs to control and eliminate *T. solium*.

## Introduction

Cysticercosis is a neglected tropical disease (NTD) that exacts a substantial health and economic burden in low-income countries. The global burden of cysticercosis includes approximately 5 million people with neurocysticercosis (NCC) or epilepsy due to NCC [1] and hundreds of millions of dollars in annual livestock losses from discarded pork [2].

Although global eradication of *T. solium* transmission is unlikely in the short-term, local control or elimination is now possible [3,4] due to the availability of new tools that can be deployed to interrupt transmission. These include effective treatment of taeniasis [5,6] and porcine cysticercosis [7,8], improved diagnostic tests [9,10], and a vaccine to prevent pig infection [11,12]. In 2012, shortly after the success of a large-scale elimination demonstration in Peru that effectively implemented many of these tools [3], the World Health Organization (WHO) declared ambitious targets for global control and elimination of *T. solium*. They called for validated control strategies to be identified by 2015, and for these strategies to be scaled and implemented in several countries by 2020 [13]. Due, in part, to the high cost and lengthy

time required to conduct prospective trials, optimal strategies for *T. solium* control have not yet been identified, and the 2020 targets for large-scale implementation have not been met. More recently, WHO called for transmission models to be deployed to optimize control strategies and reach agreed-upon targets for control [14], a goal that was recently reinforced in the WHO's 2030 goals for *T. solium* control [15].

To this end, six models of *T. solium* transmission have been published to date that attempt to fill this gap [16–21]. Despite preliminary use of these models to compare and contrast available control strategies, there are significant concerns about the validity of these models that must be addressed before reliable policy recommendations can be made. The most important deficit of existing *T. solium* models is that none have been validated against data from prospective trials carried out in the field. Comparison of model predictions with observed outcomes is a critical step for ensuring the validity of a model, and up until recently, a lack of data from prospective trials has prevented such validation.

Beyond a lack of validation, prior models have not incorporated certain features of *T. solium* transmission that may be important for model accuracy. First, no prior model has a spatially explicit structure, meaning that pigs and humans are assumed to mix homogenously to transmit the parasite, a limitation that was highlighted in a recent systematic review of available *T. solium* models [22], and in a report outlining key modeling improvements needed to meet the WHO 2030 goals [15]. Spatially clustered transmission patterns have been well-documented in endemic villages [23–26], and spatially targeted control strategies (e.g., "ring strategy") have shown success in field trials [27]. Second, the lack of an open population structure that allows for human travel and migration is a key limitation of existing *T. solium* models. The lack of such a structure prevents existing models from capturing rebounds in transmission that may occur after elimination has been achieved due to movement of infected people or pigs in to an area with successful elimination [28,29]. Together, these design features may lead existing models to over-estimate intervention effectiveness and over-estimate the likelihood of achieving control and elimination targets [30,31].

In order to address the above limitations, we developed a novel agent-based model (ABM) called "CystiAgent" [21]. Notably, CystiAgent includes a spatial structure and open population, which allows it to represent key aspects of the *T. solium* life-cycle, and facilitates the evaluation of spatially targeted control interventions. In the current study, we present results from our validation of CystiAgent using data from the RST, a cluster-randomized trial conducted in northern Peru that included 23 villages and 6 unique control strategies: four spatially targeted ring interventions and two mass applied interventions [32]. Our objective was to evaluate the ability of CystiAgent to replicate baseline levels of transmission through model calibration and to evaluate the accuracy of the model for predicting observed reductions in transmission when control strategies were applied.

## Methods

### Model description

**Model structure.** CystiAgent is a spatially explicit ABM developed in NetLogo 6.0.4 (Northwestern University, Evanston, IL), an open-access ABM software ideal for representing spatial processes in a graphical user interface. The full model code is available at http://modelingcommons.org/browse/one_model/6268, and an in-depth model description with sensitivity analyses of model parameters was previously published by Pray et al. [21].

**Agents.** CystiAgent has two classes of agents–humans and pigs–that interact and transmit *T. solium* in a spatial environment. Humans may be infected with the adult-stage intestinal tapeworm (i.e., *T. solium* taeniasis) through consumption of infected pork, and pigs may be

infected with larval-stage metacestodes (i.e., porcine cysticercosis) through contact with *T. solium* eggs or proglottids in the environment. Exposure to eggs may cause light cyst infection (<100 cysts), and exposure to proglottid segments may lead to heavy cyst infection ($\geq 100$ cysts); either may lead to seropositivity [33–35]. The different exposure-infection pathways for heavy and light cyst infection in pigs were based on a combination of biologic plausibility and consistency with prior transmission models [18,19]. Pig infection is a categorical state variable (susceptible, light infection, heavy infection), and individual *T. solium* eggs or cysts are not agents in the model. Therefore, the cut-off value of 100 cysts between light and heavy cyst infection serves only to allow for translation of prevalence from serological studies and to fit the conceptual distinction between transmission routes for light and heavy infection, but is not numerically represented in the model. Human cysticercosis, including NCC or NCC-related seizure disorders, is not included in this model.

**Environment.**  The model environment consists of households in villages that are spatially distributed according to a set of input coordinates that can be assigned to represent real villages if household coordinates are known. Household characteristics such as the presence of latrines or corrals to contain pigs are assigned at baseline and the houses are populated with humans and pigs.

**Processes and flow.**  CystiAgent consists of a sequence of biological and behavioral processes that cycle continuously to propagate the *T. solium* life-cycle (Fig 1).

1. Live pig trade. Infected pigs that are due for slaughter may be sold live within the village and then slaughtered at home by the buyer, or exported out of the village. Infected pigs from external villages may also be imported into the village through the live pig trade.

2. In-house slaughter and pork sale. Infected pigs may be slaughtered by their owners. The resulting pork meat may then be consumed at home and/or sold to other households.

3. Pork consumption. When consumed pork is infected with *T. solium* cysts, all members of the consuming households are exposed to potential tapeworm infection.

4. Human infection. Humans who consume infected pork may acquire a tapeworm infection (or remain in a susceptible state), a stochastic process determined by two locally calibrated tuning parameters (one each for consuming lightly and heavily infected pork). The intestinal tapeworm reaches maturity after 8 weeks [36], and begins expelling infectious eggs at that time. Tapeworm infections naturally clear after a stochastically determined duration [36,37].

5. Human travel. Humans that are designated as travelers leave the community at regular intervals, may contract tapeworm infections through consumption of infected pork while traveling to other endemic areas, return to the village after travel and resume contamination of their environment.

6. Open defecation. Human tapeworm carriers that do not own or use a latrine release *T. solium* eggs and proglottid segments into the environment surrounding their household location. When tapeworm infections clear, humans stop releasing proglottid segments, but contamination of the environment with eggs persists until the eggs naturally degrade according to an exponential decay function [38].

7. Pig roaming and foraging. Pigs that are designated as free-roaming (i.e., not contained in corrals) are exposed to *T. solium* proglottids and eggs that are present in their home-range areas.

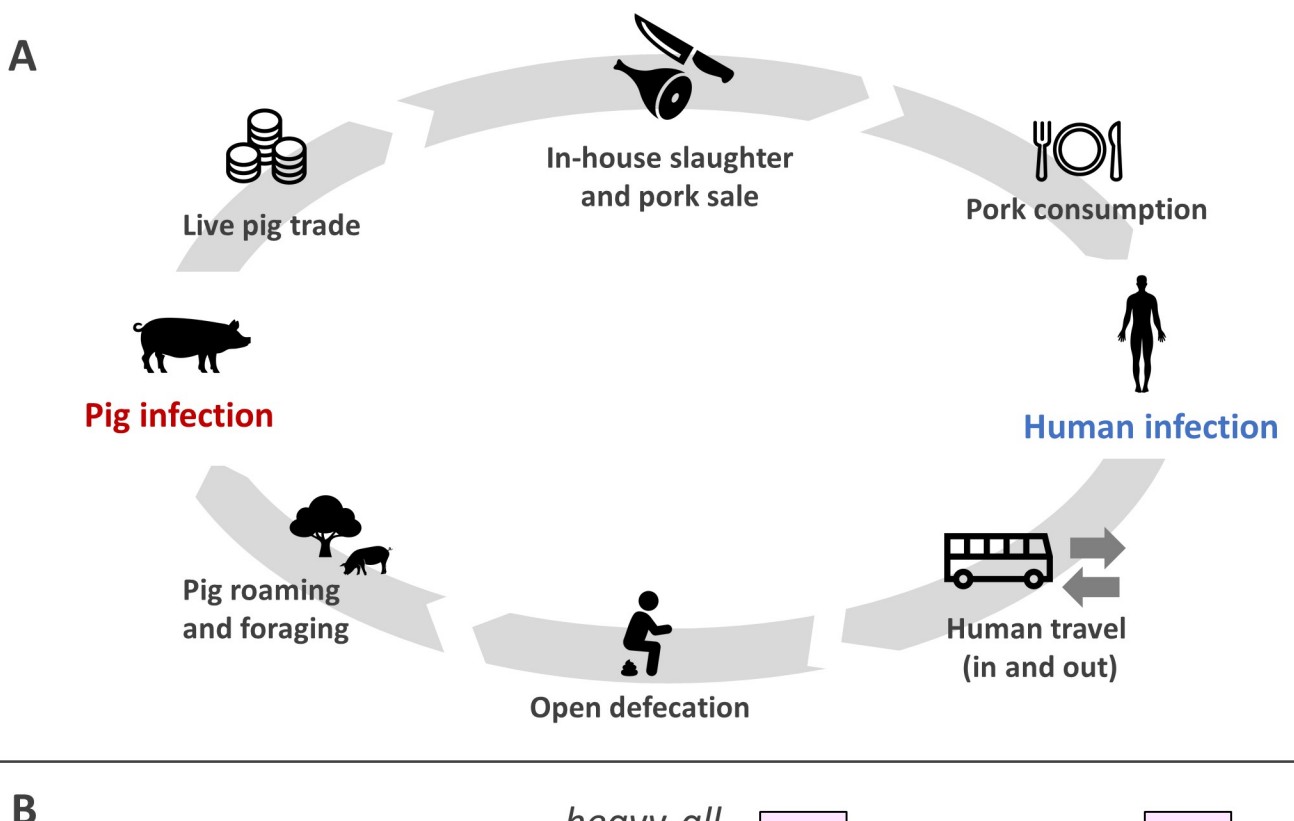

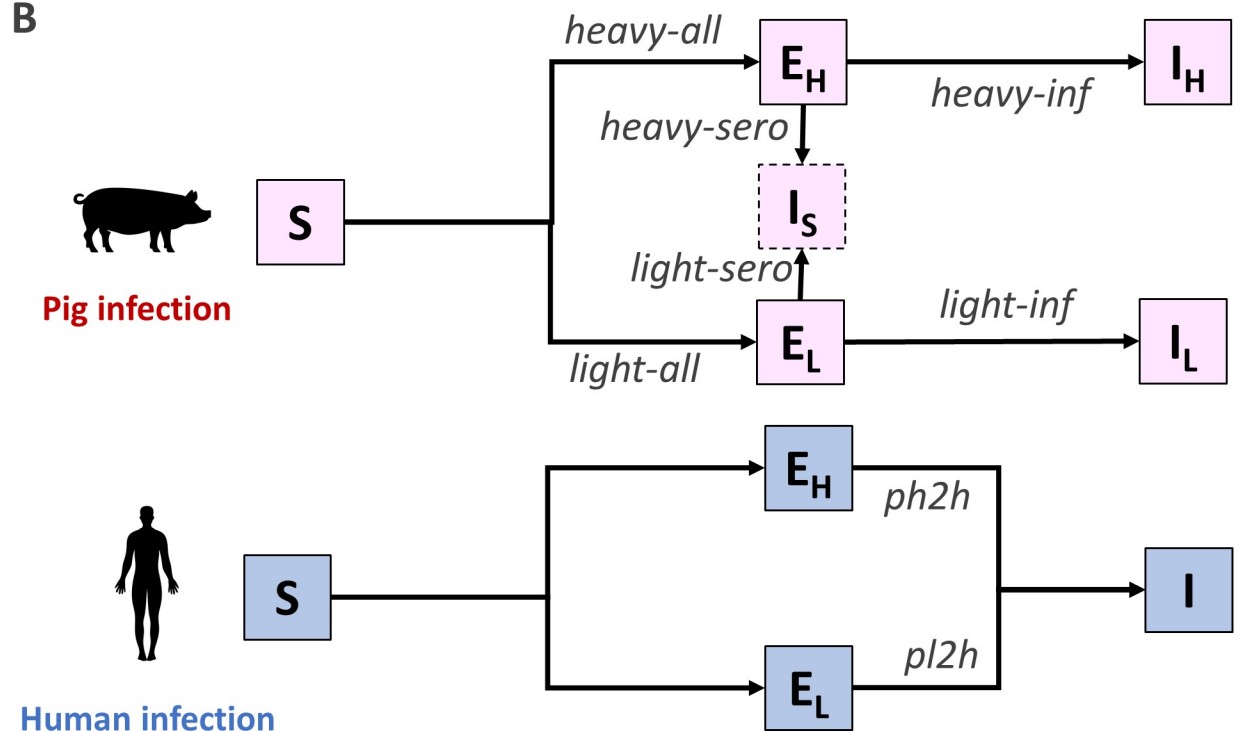

**Fig 1. Transmission of *Taenia solium* in CystiAgent.** Panel A (top): 1) Live pig trade: infected (live) pigs are sold to other households inside and outside the village, and imported from outside; 2) in-house slaughter and sale of pork to other households 3) pork consumption: potentially infected pork is consumed by household members; 4) human infection (see panel B for details); 5) human travel: humans travel to other endemic villages where they may acquire tapeworm infections through pork consumption; 6) open defecation: humans may practice open outdoor defecation; 7) pig roaming and foraging: free-roaming pigs consume potentially infectious eggs present in the environment; 8) pig infection (see panel B for details). Panel B (bottom): Susceptible pigs (S) may be heavily exposed ($E_H$) by consuming *T. solium* proglottid segments in the environment or through the stochastic parameter *heavy-all*, which is a probability of exposure applied to all pigs, or be lightly exposed ($E_L$) by consuming

*T. solium* eggs in the environment or through *light-all*. Pigs with heavy or light exposure may develop heavy ($I_H$) or light ($I_L$) cyst infection through the stochastic tuning parameters *light-inf* and *heavy-inf*. Pigs with heavy or light exposure may become seropositive ($I_S$) based on the stochastic tuning parameters *light-sero* and *heavy-sero*. Susceptible humans (S) may be heavily exposed ($E_H$) or lightly ($E_L$) exposed to cysts after eating infected pork from a heavily or lightly infected pig. Consumption of heavily or lightly infected pork cause tapeworm infection (I) in humans based on the stochastic tuning parameters *ph2h* and *pl2h*, respectively. Icons provided by Noun Project and created by Pariphat Sinma, Ikopah, Dennis, Berkah, Giorgiana Ionescu, Nabilauzwa and Alexsandr Vector.

8. Pig infection. Pigs that are exposed to proglottid segments may develop heavy cyst infection, while pigs exposed to eggs in the environment may develop light cyst infection. Either may result in seropositivity. Free-roaming pigs are exposed to an additional risk of exposure that is proportional to the number of tapeworm carriers in the village and naïve to the pig's location. This represents exposure to eggs that results from roaming and consumption of human feces from open defecation sites outside of the home area. Progression of *T. solium* exposure (eggs or proglottids) to seropositivity or cyst infection (or return to a susceptible state) is a stochastic process determined by a set of four locally calibrated tuning parameters (see below). Mature cyst infection occurs immediately after exposure without a pre-patent maturation period. Pigs that are lightly infected may progress to heavy cyst infection if exposure to proglottid segments occurs; otherwise, infected or seropositive pigs remain in that state until slaughter, with no option for recovery unless a treatment intervention is applied.

## Parameters

Each of the above processes is defined mathematically by a probability distribution and corresponding parameter(s) (Table 1). The majority of parameters were derived from observational studies conducted in northern Peru [21,32], but some were based on expert opinion and literature review. Model parameters, source data, and in-depth sensitivity analyses on model parameterization are described in detail in Pray et al [21].

## Fitting

Apart from the biological and behavioral parameters included in the model, CystiAgent utilizes a set of tuning parameters to adjust the probabilities of infection to match the observed prevalence of human and pig infection in each trial village. To do this, the model includes eight tuning parameters that represent different probabilities of exposure or infection for pigs or humans. Since these parameters represent complex sequences of unknown probabilities and cannot be determined through primary data collection or literature review, we adopted an approximate Bayesian computation (ABC) algorithm [40] to computationally derive their values for this analysis (this is described in detail below).

## Model validation—ring strategy trial

**Trial design.** The RST was a cluster-randomized trial comparing six unique population-level interventions designed to control *T. solium* transmission. The trial was carried out in 23 villages in the Piura region of northern Peru over a period of 24 months (2015–2017) [32]. Trial villages were randomly divided into six arms each receiving unique intervention designs (Table 2). Four of the six arms received some variation of ring strategy. This is an approach that targets anthelminthic treatment to humans and pigs that live within 100 meters of pigs found to be heavily infected through non-invasive tongue palpation [27,41]. Between the four ring-strategy arms, the approach varied based on the intervention applied within 100-meter rings. For humans, the two options included presumptive treatment of human taeniasis with

**Table 1. CystiAgent model parameters.**

| Parameter Description | Parameter Name | Distribution | Value (range, if applicable) | Source |
|---|---|---|---|---|
| **Village input features (village-specific)*** | | | | |
| Humans per household | humans-per-hh | Poisson | 3.8 (3.32–4.94) | RST |
| Proportion of households raising pigs | prop-pig-owners | Binomial | 0.49 (0.25–0.75) | RST |
| Pigs per pig-raising household | pigs-per-hh | Exponential | 2.43 (1.74–4.21) | RST |
| Corral prevalence among pig-owner households | prop-corrals | Binomial | 0.50 (0.23–0.92) | RST |
| Latrine prevalence | prop-latrines | Binomial | 0.64 (0.19–0.97) | RST |
| **Live pig trade** | | | | |
| Proportion of pigs sold prior to slaughter | pigs-sold | Binomial | 0.51 | HH |
| Proportion of sold pigs exported | pigs-exported | Binomial | 0.73 | HH |
| Rate of pigs imported from endemic areas (imports / pig / week) | pig-import-rate | Constant | 0.00105 | HH |
| Prevalence of cyst infection among imports | import-prev | Binomial | 0.134 | HH |
| Proportion of infected imported pigs with light cyst burden | light-to-heavy | Binomial | 0.76 | HH |
| **In-house slaughter and pork sale** | | | | |
| Pig slaughter age (months) | slaughter-age | Log-normal† | Log-mean = 2.279 (median = 9.8 months); Log-SD = 0.515 | RST |
| Proportion of pork consumed by owner | hh-only-pork | Binomial | 0.40 | HH |
| Proportion of pork sold after slaughter | sold-pork | Binomial | 0.12 | HH |
| Proportion of shared pork eaten by owner | shared-pork-hh | Binomial | 0.8 | HH |
| **Human infection** | | | | |
| Incubation time to reach tapeworm maturity | tn-incubation | Constant | 8 weeks | [36] |
| Tapeworm lifespan (years) | tn-lifespan | Normal† | Mean = 2 years SD = 0.96 years | [36,37] |
| **Human travel** | | | | |
| Proportion of households with a frequent traveler | traveler-prop | Binomial | 0.42 | HH |
| Frequency of travel to other endemic areas (every X weeks) | travel-freq | Constant | 8 weeks | HH |
| Duration of travel | travel-duration | Exponential† | 1.75 weeks | HH |
| Incidence of *T. solium* taeniasis during travel (risk / person / week) | travel-incidence | Constant | 0.00023 | [24] |
| **Open defecation** | | | | |
| Latrine-use (prop. of households that "always" use latrine) | latrine-use | Binomial | 0.25 | [39] |
| Radius of environmental contamination (meters from home) | cont-radius | Log-normal† | Log-mean = 3.27 (median = 26 meters) Log-SD = 0.547 | [39] |
| Rate of egg decay in environment (mean survival duration) | decay-mean | Exponential | 8 weeks | [38] |
| **Pig roaming and foraging** | | | | |
| Proportion of pig households with corrals that "always" corral pigs | corral-always | Binomial | 0.05 | [39] |
| Proportion of pig households with corrals that "sometimes" corral pigs | corral-sometimes | Binomial | 0.57 | [39] |
| Proportion of pigs in "sometimes"-corral-households that are corralled | prop-corral-some | Binomial | 0.32 | [39] |
| Radius of pig roaming "home-range" (meters from home) | home-range | Log-normal† | Log-mean = 3.79 (median = 44 meters) Log-SD = 0.552 | [39] |
| **Tuning parameters (village-specific)*** | | | | |
| Probability of human taeniasis upon slaughter of lightly infected pig | pl2h | Binomial | 0.00435 (0.0023–0.0099) | Calibration |
| Probability of human taeniasis upon slaughter of heavily infected pig | ph2h | Binomial | 0.00473 (0.00314–0.0075) | Calibration |
| Probability of exposure to *T. solium* eggs per human with taeniasis | light-all§ | Binomial | 0.00944 (0.00445–0.0141) | Calibration |
| Probability of exposure to *T. solium* proglottids per human with taeniasis | heavy-all§ | Binomial | 0.0161 (0.0123–0.0209) | Calibration |

(*Continued*)

**Table 1.** (Continued)

| Parameter Description | Parameter Name | Distribution | Value (range, if applicable) | Source |
|---|---|---|---|---|
| Probability of light cyst infection upon contact with to *T. solium* eggs | light-inf | Binomial | 0.286 (0.0137–0.0582) | Calibration |
| Probability of heavy cyst infection upon contact with to *T. solium* proglottids | heavy-inf | Binomial | 0.00654 (0.0041–0.0116) | Calibration |
| Probability of pig seropositivity upon exposure to *T. solium* eggs | light-sero | Binomial | 0.316 (0.095–0.67) | Calibration |
| Probability of pig seropositivity upon exposure to *T. solium* proglottids | heavy-sero | Binomial | 0.286 (0.031–0.82) | Calibration |

* Village input values were measured and applied separately in each village; tuning parameters were calibrated separately for each village; parameter values in these categories are displayed in this table as the mean and range among villages included in the analysis

† Values for these five parameters (*slaughter-age*, *tn-lifepan*, *travel-duration*, *cont-radius*, and *home-range*) were applied at the individual pig, human, or household level by randomly assigning a value from the specified statistical distribution (all other parameter values were assigned at the village level).

§ Exposure probabilities ("light-all" and "heavy-all", $x$) scaled to the current number of tapeworm carriers (HT) according to $1 - (1-x)^{HT}$

Abbreviations: RST = Ring Strategy Trial; HH = Household Survey; SD = Standard Deviation

two doses of oral niclosamide (NSM), or stool screening for taeniasis with the enzyme-linked immunosorbent assay for copro-antigen detection (CoAg-ELISA) and repeated follow-up testing and treatment with oral NSM until cure. Some villages in each of these categories received treatment of pigs with a single oral dose of oxfendazole (OFZ), and others had no pig treatment interventions. All ring interventions were administered in trial villages every 4 months throughout the 2-year trial-period. The remaining two trial arms received mass treatment of humans with a single oral dose of NSM every 6 months with and without the addition of porcine mass treatment every 4 months. Eligibility criteria for RST included all humans two years of age or older, and all pigs greater than two months of age and not pregnant.

**Table 2. Summary of interventions and populations in Ring Strategy Trial–Peru, 2015–2017.**

| Strategy | Interventions | Population |
|---|---|---|
| **Ring treatment** | Pig tongue screening, human treatment in rings (q4 months, 7x) | 2 villages* (~1200 humans, ~400 pigs) |
| **Ring treatment w/ pig treatment** | Pig tongue screening, human and pig treatment in rings (q4 months, 7x) | 4 villages (~1600 humans, 500 pigs) |
| **Ring screening** | Pig tongue screening, human screen-and-treat in rings (q4 months, 7x) | 4 villages (~1500 humans, 600 pigs) |
| **Ring screening w/ pig treatment** | Pig tongue screening, human screen-and-treat, pig treat in rings (q4 months, 7x) | 4 villages (~1500 humans, 400 pigs) |
| **Human mass treatment** | Human MDA (q6 months, 5x) | 4 villages (~1300 humans, ~400 pigs) |
| **Human and pig mass treatment** | Human and pig MDA (q6 months, 5x) | 3 villages (~1400 humans, ~500 pigs) |
| **TOTAL** | | **21 villages (~19000 humans, ~8000 pigs)** |

*2 villages excluded from ring treatment due to small size and lack of observed transmission

**Abbreviations:** MDA = Mass drug administration, q = frequency; x = repetitions.

**Trial outcomes.** Serum samples from all eligible pigs in the trial communities were collected every four months over the 24-month trial period (seven rounds total) and tested with the enzyme-linked immune-electro transfer blot (EITB). We considered a pig to be seropositive if two or more of the bands on the EITB assay were reactive [42]. Because necroscopic examination of pigs to assess true cyst infection was not conducted in the trial, we estimated the prevalence of porcine cysticercosis at baseline and trial-end using EITB serology and conversions from prior necropsy studies in the region. EITB serology is highly sensitive but non-specific for identifying pigs with active larval cyst infection (seropositivity without infection may be due to prior resolved cyst infection, exposure without established infection, or cross-reactivity), thus conversion is needed account for EITB positive pigs without active cyst infections [33,42]. We estimated that a total of 43.9% of seropositive pigs would be infected with larval cysts (31.1% of seropositive pigs with a light cyst burden of < 100 cysts, and 12.8% with a heavy cyst burden of ≥100 cysts), figures that were based on the average of two large necropsy studies conducted in Peru [34,35]. In addition to estimating the prevalence of cyst infection at baseline and trial end, we measured the cumulative incidence of EITB seroconversion for each of four-month intervals throughout the trial. This measure of seroincidence represented the proportion of new or previously seronegative pigs that became seropositive during the previous four-month period.

For human taeniasis, all human participants were offered NSM at the conclusion of the trial, and post-treatment stool samples were analyzed with CoAg-ELISA to evaluate the final-round prevalence of human taeniasis. Baseline prevalence of human taeniasis, however, was not directly measured in the RST. We estimated this value using a regression equation that was developed from a prior cross-sectional study of human taeniasis and porcine cysticercosis in the region [24]. For this, seroprevalence of (2+ EITB bands) in pigs was used a predictor for the log-prevalence of human taeniasis. A complete description of statistical methods for outcome estimation and other validation settings are included in S1 Appendix and village-level RST outcome are available in S1 Data.

**Model validation.** Validation attempts were performed individually for each village in the RST. Two of the 23 villages were excluded from validation because they were either too small to achieve stable transmission (<10 pigs in the village) or had no heavily infected pigs identified at baseline, meaning that no ring interventions were applied during RST. This left 21 villages for model validation. For each village, a unique set of household coordinates and input characteristics were applied based on RST census data and the model was fit with a unique set of tuning parameters generated by ABC calibration. Other than these village-specific settings, the same model structure and parameter values were used in all validation attempts. After calibration was performed, factors influencing calibration accuracy across villages were assessed using multivariable linear regression with relative errors at baseline as the dependent variable.

**Calibration of model tuning parameters.** We used an ABC method to estimate values for the six tuning parameters that define the probabilities of human and pig infection in the model (*pl2h*, *ph2h*, *light-inf*, *heavy-inf*, *light-all*, and *heavy-all*; see Table 1 for details) and two serological parameters (*light-sero* and *heavy-sero*) defining the probability of antibody response after exposure for pigs. A unique set of tuning parameters was estimated for each of the 21 villages. Our ABC method followed a two-stage sequential rejection sampling approach [40,43,44]. We used a Sobol' quasi-random sequence [45,46] to sample 5,000 values from a uniform prior distributions (0 to 1) of each parameter. For each combination of parameter values, we ran the model though 1000 weeks without interventions and recorded the average prevalence of taeniasis (humans), light and heavy cyst infection (pigs) and seroincidence (pigs) across the simulation period. The "sensitivity" package in R was then used to calculate the Euclidean distance [47,48] between summary statistics and the baseline prevalence observed in

the field data. We selected the top 1% of model runs that minimized the Euclidean distance and extracted posterior distributions from the selected parameter sets. With the new generated posterior distribution, we repeated the rejection sampling algorithm with 10,000 tuning parameters sets and selected our final parameter values based on the median values of the new posterior distributions produced for each parameter in this final step. All model simulations for calibration and validation were executed in R (The R Foundation for Statistical Computing, Vienna, Austria) and NetLogo on the Amazon Web Service EC2 cloud computing platform (aws.amazon.com, Seattle, WA). Model simulations were distributed across multiple 72-core parallel processors using the "parallel" R-package [49] and executed on the EC2 cloud using the R-Studio Shiny server [50].

**Simulation of control strategies.** For model validation, all simulations began with a 1000-week burn-in period, followed by the corresponding intervention sequence that was applied in the RST, and 100-week post-intervention observation period. Intervention sequences were repeated 1000 times per village for validation. Participation in interventions among humans and pigs was applied separately in each village and reflected the proportions that were observed in that village during the RST. The sensitivity and specificity of the CoAg-ELISA for detecting *T. solium* taeniasis was set to 96.4% and 100%, respectively based on the method described by Guezala et al. [9], and the efficacy of NSM for treatment of human taeniasis was set at 76.6% for one dose, 86.6% for two doses, and 93.3% for post-screening follow-up. Treatment of pigs with OFZ was assumed to have an efficacy of 100%. Detailed model settings used for validation are available in S1 Appendix and participation rates used for each village are available in S1 Data.

**Evaluation of model accuracy.** For each RST village, we compared the model-predicted prevalence of human taeniasis, porcine cysticercosis, and pig seroincidence with the values observed in the trial (or values estimated from trial observations). For graphical comparison, the plots of model-predicted outcomes show the absolute and relative error of the median of the 1000 model runs compared to the values observed in the trial or generated from trial observations. The relative error and absolute errors between an observed $P_{obs}$ and the median simulated prevalence $P_{sim}$ is defined as follows:

$$Absolute\ error = P_{sim} - P_{obs} \quad Relative\ error = \frac{P_{sim} - P_{obs}}{P_{obs}}$$

## Results

### Baseline calibration for ring strategy trial

Among the 21 villages evaluated in the RST, the median observed prevalence of human taeniasis and porcine cysticercosis at baseline were 2.2% (range: 0.9% to 3.8%) and 19.2% (range: 6.3% to 27%), respectively. Median porcine seroincidence per 4-month period at baseline was 44.2% (range: 5.3% to 78%).

The ABC calibration step allowed CystiAgent to accurately replicate observed baseline prevalence for all villages and disease outcomes (Fig 2). The median absolute difference between simulated and observed taeniasis prevalence at baseline was +0.09 percentage points (range: -0.20 to +0.63 percentage points), corresponding to a median relative error of +4.9% (range: -18.5% to +33.3%). For porcine cysticercosis, the median absolute difference between simulated and observed prevalence at baseline was +0.58 percentage points (range: -1.6 to +3.1 percentage points), with a median relative error of +3.8% (range: -9.3% to +22.8%). For porcine seroincidence, median absolute differences and relative errors between simulated and observed at baseline were +1.8 percentage points (range: -3.4 to 5.3 percentage points) and +3.2% (range: -4.7% to +27.9%), respectively. Overall, calibration accuracy was high across all

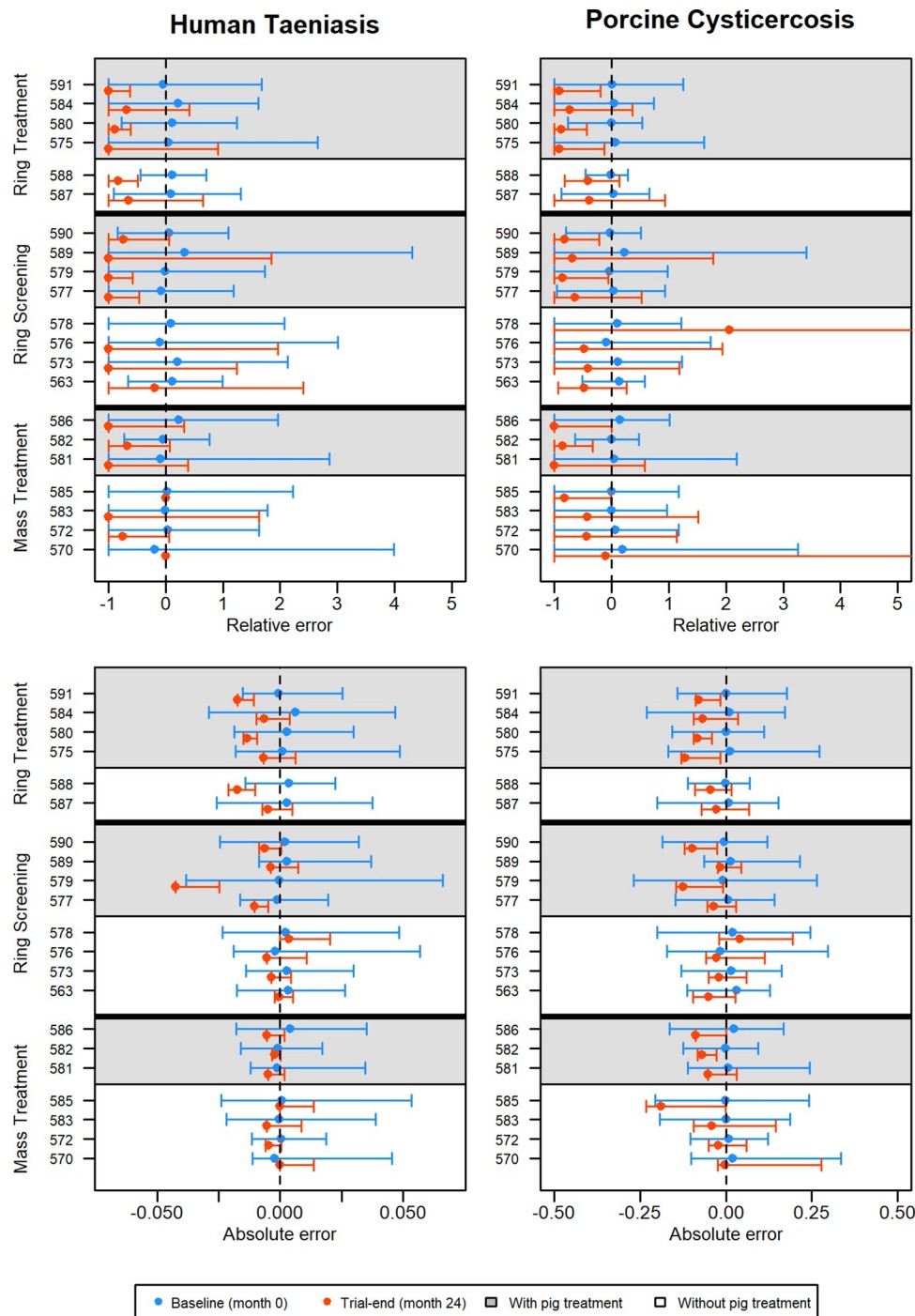

**Fig 2. Relative (top) and absolute (bottom) errors during validation of CystiAgent against results from the Ring Strategy Trial for human taeniasis (left) and porcine cysticercosis (right) at baseline (month 0) and trial-end (month 24) (n = 21 villages).** For ring treatment (n = 6 villages), ring screening (n = 8 villages) and mass treatment (n = 7 villages villages), interventions with (white) and without (grey) pig treatment are designated by the background color of the plot area. Three villages (570, 578, and 585) had no human taeniasis identified at trial-end and relative errors were not able to be calculated.

villages, and no village factor in the model, including number of households, use of corrals and use of latrines were associated with an improvement or decline in the accuracy of baseline calibration after evaluating calibration errors in a multivariable linear regression model.

## Accuracy of intervention effects

When RST intervention strategies were simulated in CystiAgent, the model overestimated the intervention effects for taeniasis and porcine cysticercosis in the majority of villages. The median prevalence of taeniasis observed at the conclusion of the RST was 0.55% (range: 0 to 4.3%), with elimination of taeniasis by trial-end in 3 of the 21 villages. CystiAgent predicted elimination of taeniasis at trial-end in 12 of 21 villages and median prevalence across all villages of 0% (range: 0 to 0.37%), which was 0.53 percentage points less (range: 4.3 to +0.36 percentage points) than the observed prevalence at trial-end.

The median prevalence of porcine cysticercosis observed at the conclusion of the RST was 8.8% (range: 2.0 to 23.3%). CystiAgent predicted a trial-end median prevalence of 2.2% (range: 0% to 6.4%), which was a median of 5.0 percentage points less (range: 18.9 to +4.1 percentage points) than the observed prevalence of porcine cysticercosis at trial-end. None of the villages achieved complete elimination of taeniasis and porcine cysticercosis by the end of the trial; however, CystiAgent predicted *T. solium* elimination in two of the 21 villages at the end of the trial, and six villages by the end of the 100-week post-intervention period.

Between the three primary intervention types compared in RST (ring screening, ring treatment, and mass treatment), CystiAgent predicted similar and substantial reductions in taeniasis and porcine cysticercosis by trial-end (Fig 3). This differed from observed results of the trial, which found that ring-screening and mass-treatment led to larger reductions in prevalence compared to ring-treatment, although all three intervention types were effective control strategies against human taeniasis and porcine cysticercosis. Prediction errors for taeniasis and porcine cysticercosis at trial-end were greatest for ring treatment (-1.0 and -7.4 percentage points for taeniasis and porcine cysticercosis, respectively), and less for ring screening (-0.47 and -3.1 percentage points), and mass treatment (-0.45 and -5.2 percentage points).

In villages where antihelminthic treatment of pigs was added to each of the three primary intervention types, CystiAgent predicted more dramatic reductions in the prevalence of porcine cysticercosis compared to interventions with human screening or treatment alone. The predicted effect of pig treatment on human taeniasis prevalence was minimal. Graphs of model predictions versus RST outcomes split by interventions with and without pig treatment are available in S1 Fig. Despite model predictions, an additive effect of pig treatment was not observed in RST, which led to increased prediction errors in villages where pig treatment was applied.

The accuracy of CystiAgent with respect to porcine seroincidence as a measured outcome in RST was similar to other outcomes, with the model slightly overestimating the intervention effect in most villages. Across all 21 RST villages, the median observed porcine seroincidence in the final 4-month study interval was 12.6% (range: 3.3% to 45%), which corresponds to a median decline of 24.8 percentage points (range: 1.6 to 71 percentage points) from baseline. In CystiAgent, the median simulated seroincidence in the final 4-month study interval was 6.1% (range: 0% to 25.7%), which corresponds to a median absolute difference of 6.6 percentage points lower (range: -36.2 to +18.7 percentage points) than the observed seroincidence during that interval.

Comparing the accuracy of seroincidence simulations across intervention types in RST, all three overestimated intervention effects by a similar degree by trial-end (Fig 4). CystiAgent predicted seroincidences during final 4-month study interval that were a median of 6.2, 7.0, and 6.6 percentage points less than observed seroincidence in ring screening, ring treatment, and mass treatment villages, respectively. Despite this consistency at trial-end, model accuracy

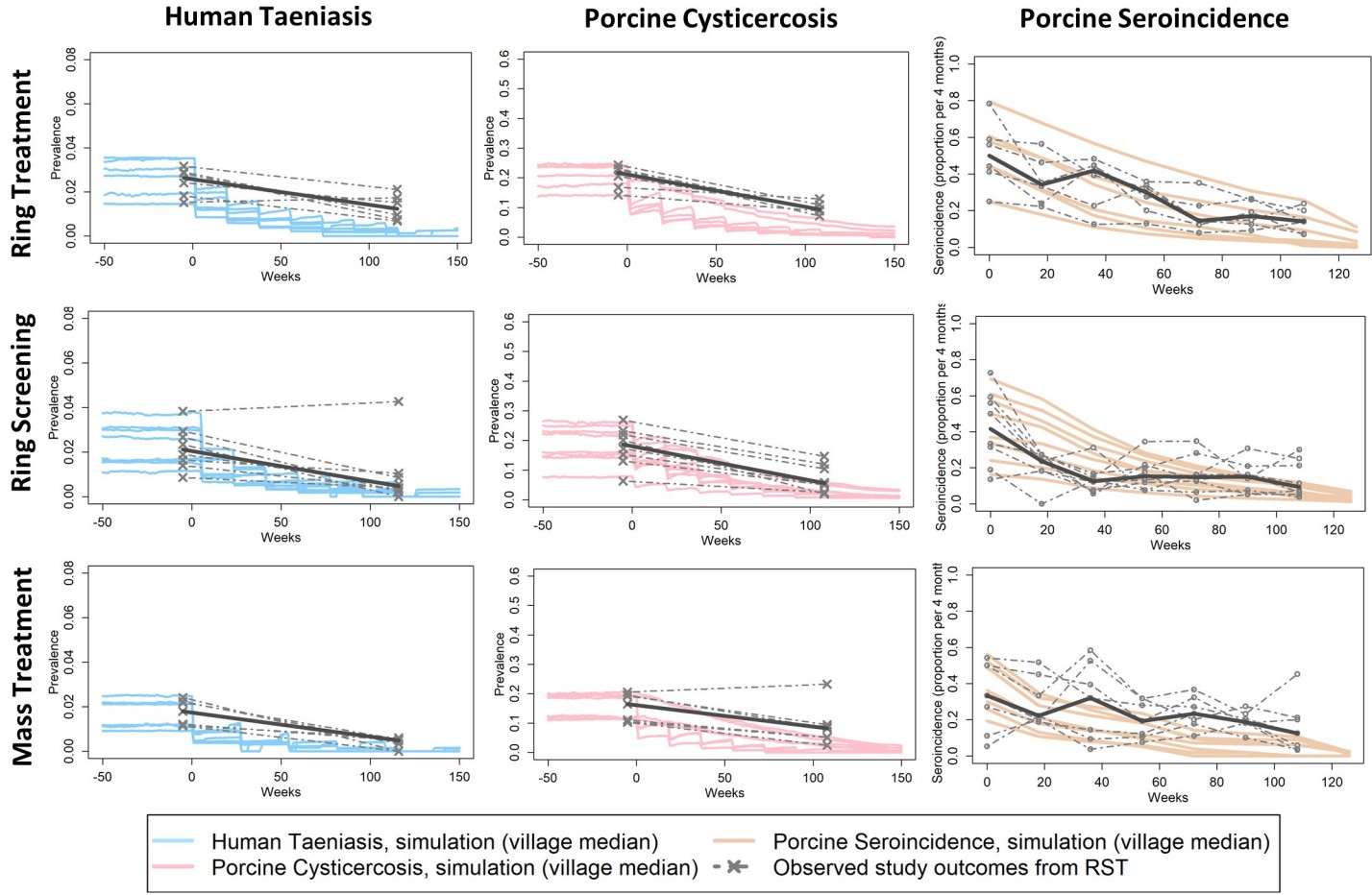

**Fig 3. Simulation outcomes versus observed rates of human taeniasis, porcine cysticercosis and porcine seroincidence in the Ring Strategy Trial, by intervention type (n = 21 villages).** Plots display village-specific median simulated outcomes in CystiAgent across 1000 simulations per village, and observed outcomes from RST (median value within intervention type in bold). For ring treatment (n = 6 villages), ring screening (n = 8 villages) and mass treatment (n = 7 villages villages), intervention and without pig treatment are combined. See S1 Fig for sub-arm comparisons of villages with and without the addition of pig treatment.

varied by intervention type at mid-study time-points 2 and 3 (months 4 and 8 of RST), when CystiAgent failed to replicate a steep early decline in seroincidence that was observed only in ring screening villages.

## Discussion

Development and validation of a *T. solium* transmission model was identified as a key goal in WHO's 2014 framework for intensified control [14], yet none of the currently available models have been validated against data from field trials of *T. solium* control interventions. In this research, we developed a novel transmission model for *T. solium* called CystiAgent, and validated the model using data from a large cluster-randomized trial of ring and mass-applied pharmaceutical intervention conducted in northern Peru. Validation results showed that CystiAgent could be accurately calibrated to reflect baseline levels of *T. solium* transmission, but overestimated the magnitude of intervention effects in most villages. Despite this, the current validation allowed for identification of potential gaps in the model that may be addressed in future versions, and represented an important step towards delivering an accurate and validated model that can be used to answer key questions about *T. solium* transmission and control.

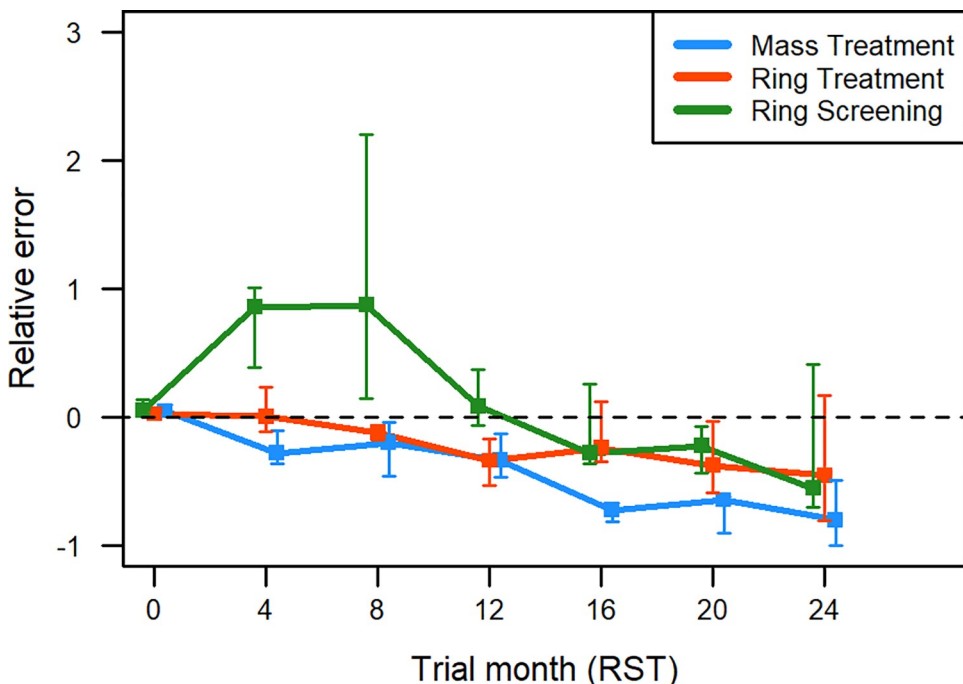

**Fig 4. Relative errors of porcine seroincidence during validation of CystiAgent against result from the Ring Strategy Trial, by intervention type–ring treatment (n = 6 villages), ring screening (n = 8 villages), and mass treatment (n = 7 villages).** Each time point represents the 4-month cumulative seroincidence across the 24-month trial period, as measured by enzyme-linked immune-electro transfer blot (EITB) in pigs. Error bars show inter-quartile range for relative errors among village in each intervention. Within each intervention type, villages with and without pig treatment are combined.

## Baseline calibration

For this validation, CystiAgent was calibrated to each study village individually using the model's eight tuning parameters and a complex ABC estimation method. Using this method, baseline calibration was highly accurate in nearly all villages (median calibration errors of +0.09 and +0.58 percentage points for taeniasis and porcine cysticercosis, respectively), which provided assurance that the model could reproduce observed levels of endemic transmission. Two villages were excluded from the study because their small population size and lack of infected pigs led to unstable transmission in the model; therefore, very small villages (<30 households) may not be suitable for CystiAgent. Estimation of tuning parameters separately for each village with ABC estimation is computationally demanding, and may not be feasible for larger scale use of CystiAgent to compare or evaluate *T. solium* control interventions. To address this limitation, updates to CystiAgent that may be considered for future versions include reducing the number of tuning parameters and fitting a single set of parameter values to prevalence levels averaged across a larger geographic region. Such adjustments would benefit from additional sensitivity analysis and validation to evaluate the impact of these changes on model performance.

## Validation of intervention strategies

Validation of CystiAgent against RST interventions showed that the model was able to execute all six intervention types (ring screening, ring treatment, and mass treatment; with and without pig treatment) and reproduce observed declines in transmission; however the model overestimated the magnitude of intervention effects in the majority of villages tested. The median

trial-end prevalence for taeniasis and porcine cysticercosis simulated in CystiAgent were 0.53 percentage points lower (0.55% observed vs. 0% simulated) and 5.0 percentage points lower (8.8% observed vs. 2.2% simulated) than the observed prevalences in RST, respectively.

There are a few possible explanations for this error that could be addressed with additional data collection to improve CystiAgent performance. First, the efficacy of NSM for curing taeniasis may have been overestimated in the model. While the applied single-dose efficacy of 76.6% has been reported elsewhere [51], variations in manufacturer formulation and shelf-life could impact the treatment efficacy. Such a difference would reduce the effectiveness of control interventions and could have contributed to the overestimated intervention effects predicted in CystiAgent. Second, heterogeneous non-participation in control intervention, particularly if concentrated among persons at high-risk for *T. solium* infections, would reduce the effectiveness of control interventions. In CystiAgent, treatment applications among simulation participants was randomly assigned among eligible individuals without accounting for systematic non-participation, which could have led to overestimation of intervention effectiveness. Third, acquired immunity and resistance to *T. solium* infection in pigs was not included in CystiAgent due to a lack of data to accurately parameterize this process. Pig immunity is highly dynamic [33] and the number of pigs with acquired immunity in the population would likely decrease as control interventions are applied due to lower levels of exposures to *T. solium* eggs in the environment, which could lead to more rapid rebounds in transmission among susceptible pigs after treatment applications. CystiAgent was not able to replicate these dynamic changes in population-level resistance among pigs, which may have caused overestimation of the intervention effects predicted by CystiAgent.

Model performance with respect to porcine seroincidence by round showed that the model predicted steady declines in seroincidence after each successive intervention application for all three intervention types. While this trajectory matched the observed patterns of decline in villages that received ring treatment and mass treatment interventions (despite overestimating the magnitude of decline), villages that received the ring screening intervention experienced a sharp initial decline in seroincidence followed by a leveling off at low levels of transmission that was not replicated in the model. The mechanism underlying this sharp decline and subsequent leveling off in ring screening arms is not definitively known. Possible explanations for not capturing this pattern in CystiAgent include a higher efficacy of NSM when used post-screening (91.8% post-screening efficacy was applied in CystiAgent), or a higher degree of spatial clustering among *T. solium*-infected humans and pigs than modeled.

## Strengths and limitations

There are a few important strengths of this validation to highlight. First, we had access to all data collected during RST, a unique opportunity that allowed for this validation to be conducted. Using these data, the model was tested independently in 21 separate villages and 6 unique intervention types, leading to a robust assessment of the model's accuracy in a variety of transmission settings. The consistency of the model's performance across all villages provides initial support for the generalizability of the model to other transmission settings. Second, the spatially explicit structure of CystiAgent allowed us to evaluate ring interventions in this validation. Ring interventions are spatially targeted strategies for *T. solium* control that have shown promising results in field studies [27,32], and are not able to be evaluated in other non-spatial *T. solium* models.

There are two main limitations of this validation study. First, baseline and trial-end measures of porcine cysticercosis prevalence were limited by the diagnostic methods used in RST. CystiAgent estimated the prevalence of true cyst infection in pigs based on EITB serology

because necroscopic examination to definitely diagnose cyst infection in pigs was not conducted in RST. EITB serology is highly sensitive but poorly specific for detecting true cyst infections [33,52]. Therefore, to approximate the prevalence of true cyst infection at baseline and trial-end, we estimated that 44% of seropositive pigs based on EITB would be infected, a figure based on two necropsy studies previously conducted in the region [34,35]. These necropsy studies were conducted in different pig populations in different regions of Peru that had not received intervention programs, and may not have the same immune responses as pigs in this trial. Also, we used the same conversion factor for EITB serology at baseline and trial-end, but the diagnostic performance of EITB may change after interventions are applied due to changing levels of immunity in the pig population. This effect was observed in a previous trial in Peru [3], and could indicate that trial-end prevalence of cyst infection in RST was lower than estimated, which would indicate that model predictions were more accurate than the measured error-rates indicated. More research is needed to determine how treatment and disease control impacts population immunity and diagnostic performance such that these features can be accounted for in future models. Secondly, parameters used for this validation were primarily sourced from a single region of northern Peru (Piura region). While the depth of data available from this region made it possible to develop and validate this detailed ABM, some key parameters may differ in other endemic areas of the world, which could impact the model's accuracy when applied to these regions. Application of CystiAgent outside of northern Peru, therefore, would benefit from local data to set appropriate values for key model parameters such as latrine use, pig roaming ranges, and pig trading, which were identified as high priority parameter groups in prior sensitivity analyses conducted with CystiAgent [21].

## Future research priorities

CystiAgent and other *T. solium* models will always be limited by uncertainty in the mechanisms and dynamics of *T. solium* transmission, which could impact model accuracy and validity. Some of the more important gaps in knowledge that may impact transmission but are not yet incorporated into CystiAgent include age-related differences in pig roaming and environmental exposure to *T. solium* [39,53,54], distribution patterns of infected pork through black market channels [55], and the possibility of vector-borne transmission of *T. solium* eggs via dung beetles and flies [56,57]. Perhaps the most important aspect of *T. solium* transmission not included in the model, however, is immunity. Due to insufficient knowledge of the mechanisms of immunity, resistance, and susceptibility to *T. solium* infection in humans and pigs, we were not able to incorporate these features into the model. As described above, this may have contributed to the observed overestimation of intervention effects. Since probabilities of infection in the model (i.e., tuning parameters) were estimated using levels of transmission observed at baseline, the model is not able to account for changes in susceptibility or resistance that may occur in response to control pressure when an intervention is applied. This feature could be added to the model if appropriate data from experimental and/or field studies becomes available. While these and many other factors are not yet features of CystiAgent, a key advantage of its ABM structure is that it has the flexibility to incorporate these novel features when data become available, and can serve as an accessible platform to develop and test hypotheses about *T. solium* transmission dynamics.

## Conclusion

In this research, we developed and validated a novel transmission model for *T. solium* called CystiAgent. In this first large-scale validation of a *T. solium* transmission model, CystiAgent was able to simulate levels of transmission observed at baseline in the validation villages, and

adequately replicate the effects of control interventions in the majority of villages. Overall, the model overestimated intervention effects in many of the villages tested. These imperfect results, however, represent important data-points that can be used to adjust and improve future versions of the model. Moving forward we will continue to test and improve the CystiAgent model using data available from interventions in Peru, and evaluate the generalizability of the model through validation against data from other endemic regions. Ultimately, we aim to use a final validated version of CystiAgent to evaluate available strategies for *T. solium* control and deliver evidence-based recommendations that will meet the need for validated strategies emphasized by WHO.

## Supporting information

**S1 Appendix. Additional methods for comparison of CystiAgent simulations with observed results from the Ring Strategy Trial.**
(DOCX)

**S1 Data. Data file including CystiAgent simulation results, observed results from the Ring Strategy Trial, and all intervention settings for each village evaluated (n = 21 villages).**
(CSV)

**S1 Fig. Performance of CystiAgent model across six intervention strategies in RST (ring screening, ring treatment, and mass treatment; each with and without the addition of pig treatment) (n = 21 villages).** Plots display village-specific median simulated outcomes in CystiAgent across 1000 simulations per village, and observed outcomes from RST (median value within intervention type in bold). For ring treatment (n = 6 villages), ring screening (n = 8 villages) and mass treatment (n = 7 villages villages), interventions with (grey) and without (white) pig treatment are designated by the background color.
(TIF)

## Author Contributions

**Conceptualization:** Ian W. Pray, Wayne Wakeland, William K. Pan, William E. Lambert, Seth E. O'Neal.

**Data curation:** Ian W. Pray, Francesco Pizzitutti, Gabrielle Bonnet, Eloy Gonzales-Gustavson, Seth E. O'Neal.

**Formal analysis:** Ian W. Pray, Francesco Pizzitutti, Gabrielle Bonnet, Eloy Gonzales-Gustavson.

**Funding acquisition:** Armando E. Gonzalez, Hector H. Garcia, Seth E. O'Neal.

**Investigation:** Ian W. Pray.

**Methodology:** Ian W. Pray, Wayne Wakeland, William K. Pan, William E. Lambert, Seth E. O'Neal.

**Project administration:** Ian W. Pray, Armando E. Gonzalez, Hector H. Garcia, Seth E. O'Neal.

**Resources:** Armando E. Gonzalez, Hector H. Garcia, Seth E. O'Neal.

**Software:** Ian W. Pray.

**Supervision:** Wayne Wakeland, William K. Pan, William E. Lambert, Seth E. O'Neal.

**Validation:** Ian W. Pray, Francesco Pizzitutti, Gabrielle Bonnet, Eloy Gonzales-Gustavson.

**Visualization:** Ian W. Pray, Francesco Pizzitutti.

**Writing – original draft:** Ian W. Pray.

**Writing – review & editing:** Ian W. Pray, Francesco Pizzitutti, Gabrielle Bonnet, Eloy Gonzales-Gustavson, Wayne Wakeland, William K. Pan, William E. Lambert, Armando E. Gonzalez, Hector H. Garcia, Seth E. O'Neal.

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
