## [Decision Letter · Decision Letter 0]

30 Mar 2021

Dear Mr. Pray,

Thank you very much for submitting your manuscript "Validation of a spatial agent-based model for Taenia solium transmission (“CystiAgent”) against a large prospective trial of control strategies in northern Peru" for consideration at PLOS Neglected Tropical Diseases. As with all papers reviewed by the journal, your manuscript was reviewed by members of the editorial board and by several independent reviewers. In light of the reviews (below this email), we would like to invite the resubmission of a significantly-revised version that takes into account the reviewers' comments. 

We cannot make any decision about publication until we have seen the revised manuscript and your response to the reviewers' comments. Your revised manuscript is also likely to be sent to reviewers for further evaluation.

Sincerely,

Kendall McKenzie

Staff Admin

Makedonka Mitreva

Deputy Editor

Reviewer's Responses to Questions

**Key Review Criteria Required for Acceptance?**

**Methods**

-Are the objectives of the study clearly articulated with a clear testable hypothesis stated?

-Is the study design appropriate to address the stated objectives?

-Is the population clearly described and appropriate for the hypothesis being tested?

-Is the sample size sufficient to ensure adequate power to address the hypothesis being tested?

-Were correct statistical analysis used to support conclusions?

-Are there concerns about ethical or regulatory requirements being met?

Reviewer #1: (No Response)

Reviewer #2: The methods present the published CystiAgent spatially explicitly model, parameterised for Northern Peru and fitted using an ABC methodology (probability of exposure/infection "tuning" parameters) to baseline village-level prevalence. Interventions applied in the ring strategy trial (RST) are used to validate CystiAgent, by comparing the relative/absolute errors of model projections vs observed data (porcine seroincidence) or estimated outcomes (porcine cysticercosis "true" cyst prevalence or human taeniasis prevalence). The methods are well written, and appropriate to address to address the study objectives. I have a number of questions which are outlined in the "summary/general comments" section.

**Results**

-Does the analysis presented match the analysis plan?

-Are the results clearly and completely presented?

-Are the figures (Tables, Images) of sufficient quality for clarity?

Reviewer #1: (No Response)

Reviewer #2: The results address the analysis plan, are well presented (including the figures), showing the ability of CystiAgent to accurately capture the baseline endemic prevalence in each village following fitting of the "tuning" probabililty parameters. Model intervention projections were overestimated compared to observed data (porcine seroincidence) or estimated outcomes (porcine cysticercosis "true" cyst prevalence or human taeniasis prevalence), represented by the visual model projections comapred to data, and relative/absolute error plots. I have a number of questions and recommendations for improvements outlined in the "summary/general comments" section.

**Conclusions**

-Are the conclusions supported by the data presented?

-Are the limitations of analysis clearly described?

-Do the authors discuss how these data can be helpful to advance our understanding of the topic under study?

-Is public health relevance addressed?

Reviewer #1: (No Response)

Reviewer #2: The authors provide a rich discussion into the discussion section, in particular describing possible explanations for the observation of overestimation model intervention impact projections compared to data/estimated outcomes. The wider implications are described, showing the utility of using a spatially-explicit model to consider the contribution of spatial epidemiological processes and testing spatial interventions. I have outlined some key considerations and recommendations in the "summary/general comments" section.

**Editorial and Data Presentation Modifications?**

Reviewer #1: (No Response)

Reviewer #2: Please see further comments for the results section under the "Summary and General comments" section.

**Summary and General Comments**

Reviewer #1: The manuscript provides a very valuable evaluation of the CystiAgent Taenia solium disease transmission model, and generates avenues for further research. I only have some minor suggestions:

1. L418, "prevent.. from accurately replicating observed patterns": one could argue if this is really the goal of any transmission model, or if it would even be possible. To replicate observed patterns, one should have a perfect understanding of reality, while the role of simulation models is rather to better understand reality through simplification. So the lack of a spatially explicit structure, or of an open population structure, would not be a "structural deficit" per se, but rather a design choice, ie, simplification.

2. Table 1: it is not always clear how exactly the distribution and parameter should be understood. Do all distributions represent variability, and not uncertainty? For Poisson, Exponential and Binomial (assuming n=1), one parameter would indeed suffice, but for other parameters it would not. I would therefore appreciate some more info on how to interpret the uniform, log-normal and normal distributions.

3. P12L3: the exclusion of 2 villages could be seen as a possible source of bias in the validation results, and could merit some discussion. Does it mean the model is not able to deal with low prevalence settings?

4. P10L2: Would it be possible to include a reference to the manuscript in review?

5. The terms "ring strategy" and "ring strategy trial" are sometimes capitalized, especially in the beginning of the manuscript. The abbreviation "RST" is introduced on page 9 but could be introduced earlier on.

6. In the author summary, "T. solium" should be written in full at first occurrence.

Reviewer #2: Intro section:

Minor comments:

1. Pg. 3 Lines 19-20: presumably the “over 5 million people with epilepsy from NCC” figure comes from Table 1 of Ref[1]; if this is the case, the average figure is 4.85 million cases and relates to persons infected with NCC or epilepsy due to NCC, so this should be written as “approximately 5 million people with neurocysticercosis (NCC) or epilepsy due to NCC”.

2. Pg. 5 Lines 2-3 : This is not correct, existing transmission model with closed population structure would be able to capture rebound in prevalence following intervention cessation (if interventions did not achieve local elimination). The authors would be more accurate in stating existing models would not be able to capture rebound following elimination, due to movement of infected people into an area with successful elimination. 

3. Pg.5 lines 3-5: I do not think the authors can word this so strongly. These deficits *may*result in over-estimation of predictions, but really cross-validation of models on the same datasets will be required to determine differences in predictive capacity (i.e. cross-validation/comparison efforts are more common-place in modelling areas for other NTDs). 

Methods section:

1. It is not clear from the methods text, and tables 1 and Figure 1, how long pigs remain in the “exposed” state, or how this is represented in the model. Does the represent the pre-patent period in pigs (before cyst maturation?)- from Table 1 and description in the methods section text, my impression is that pigs progress immediately after exposure based on the tuning parameter light-inf or heavy-inf without including a cyst maturation period in pigs? Please can the authors make this clearer in the text and describe the way progression from exposed to infected is parameterised in the model. 

2. I am a little unclear regarding the implications of lines 10-13 on page 11; so 42.9% are identified as infected if seropositive (30.1% with light infections plus 12.8% with heavy cyst infections), does this imply that 57.1% are false-positive or antibody positive from the EITB but not harbouring active larval infection (i.e. recovered)?

3. Can the authors indicate whether the copro-antigen ELISA protocol used matched the Guezela et al. 2009 protocl (ref 4 in S1 appendix). (and). This is important as 96.4% sensitivity suggests this protocol was used, whereas sensitivity of copro-antigen ELISA from other literature estimates is generally lower (for example see the modelled estimate from Praet et al. 2013 with a sensitivity of 84.5%). Other copro-antigen protocols based on the original Allan et al. 1999 protocol) give specificity estimates of <100% (not species specific), whereas the Guezela et al. 2009 protocol gave a specificity of 100% (so presumably the authors are assuming 100% specificty, although it is not clear?)- more discussion on the different protocols for copro-antigen ELISA tests can be found in Lightowlers et al. 2016. 

Praet N, Verweij JJ, Mwape KE, Phiri IK, Muma JB, Zulu G, van Lieshout L, Rodriguez-Hidalgo R, Benitez-Ortiz W, Dorny P, Gabriël S. Bayesian modelling to estimate the test characteristics of coprology, coproantigen ELISA and a novel real-time PCR for the diagnosis of taeniasis. Trop Med Int Health. 2013 May;18(5):608-14.

Lightowlers, M W et al. “Monitoring the outcomes of interventions against Taenia solium: options and suggestions.” Parasite immunology vol. 38,3 (2016): 158-69. doi:10.1111/pim.12291

Minor: 

1. Can the authors provide a reference and explain the rationale for the cut-off for light infections with <100 cysts and heavy infections with > 100 cysts (pg.6 lines 6-7). 

2. Can the authors please state in the table which parameters refer to light-sero and heavy-sero from the description in the text (lines 11-12, page 12) - presumably this is the last row in table 1, but this should be made explicit.

3. Please explicitly state that min/max and other quantities are available to specify statistical distributions in the Pray et al. 2020 CystiAgent publication (and that these are the same as presented in this manuscript), as this is not clear. 

Results section:

1. Is there a reason for not estimating the prevalence of cyst infection from the EITB seroincidence data every 4 months, so that prevalence estimates could be more closely compared to simulated dynamics on porcine cysticercosis prevalence throughout the trial (i.e. middle panel in Figure 3)? Equally, why was human taeniasis prevalence not estimated for each timepoint (from the regression model using EITB data) to compare taeniasis prevalence estimates with modelled estimates (throughout the trial)? I think the analysis would be enhanced if these estimates are presented vs. model projections in Figure 3 (rather than just comparing baseline and end-of trial estimates to modelled projections) - the relative/abolsute error plots can still just show differences at baseline vs. end of trial.

2. Where the authors state that various factors are not associated with an improvement or decline in the accuracy of baseline calibration (lines 15-17 pg.14), how was this tested - from the sensitivty analysis in the original Pray et al. 2020 CystiAgent model paper?

3. The authors state in the legend of Figures 2- 4 that the intervention with and without pig treatment are combined, is there a reason for this? I would recommend showing the model projections and observed data broken down by +/- pig treatment (at least in the supplementary information) to show whether pig treatment has a systematic impact on difference between observed data and projections between ring treatment/ring screening/mass treatment (and the relative error in figure 4). 

4. There seems to be a trial-end point missing for the human taeniasis relative error panel in figure 2 (village 576 for ring screening and village 570 for mass treatment)?

Minor:

1. Is the median porcine seroincidence a % per 4 months (lines 6 on page 14)? Can the authors explicitly state this if so i.e. 44.2% per 4 months at baseline. I recommend in Figure 3 that the seroincidence units on the y-axis (of the 3rd column/panel) should be “proportion per 4 months”. 

2. If I am understanding correctly, the median difference referenced on line 8 (for taeniasis) and 11 (porcine cysticercosis) of pg.14 relate to the median (absolute) differences? Can the authors explicitly state whether they refer to the absolute or relative differences throughout the manuscript please.

3. The abbreviations for porcine cysticercosis (PC) and taeniasis (HT) are first introduced in line 22 on page 15, however should be introduced much earlier, with abbreviations used on line 24 of page 3 (if the authors wish to use abbreviations).

4. On line 17-18 of page 16, should this read as range: -36.2 to +18.7% percentage points or 36.2 lower to 18.7% higher to improve clarity

Discussion section:

1. The authors indicate that they have used data to suggest that 44% of EITB-seropositive pigs would be infected (should this not be 43% from 30.1% light infected + 12.8% heavy infected?) – I think an improved approach would be to adjust observed prevalence data/estimates to 'informed' or 'true' prevalence with the methodology proposed by Speybroeck et al. 2013 (using a Bayesian framework, incorporating uncertainty in diagnostic sensitivity and specificity)? If evidence from previous intervention programmes suggest that the false positive rate (specificity) changes (as the authors suggest in lines 1-3 of page 21), the observed results in the RST could be adjusted based on the method proposed by Spebroeck et al. 2013 with varying specificity throughout the programme? I recommend this as an area for the authors to consider for adjusting their field data, especially if the diagnostic parameters change throughout intervention (which, as the authors suggest may change the error between simulated prevalence and prevalence estimates from the data). 

Speybroeck N, Devleesschauwer B, Joseph L, Berkvens D. Misclassification errors in prevalence estimation: Bayesian handling with care. Int J Public Health. 2013 Oct;58(5):791-5.

Minor:

1. Could the authors explain the statement that the number of pigs with acquired immunity would decrease as control interventions are applied (lines 9-10 on page 19); is this because reduction in environmental contamination of the environment (through treatment of taeniasis carriers) would result in lower exposure of pigs (and therefore development of immunity)? Does this then assume that susceptibility in the pig population is higher when pigs are subsequently exposed (following imported cases of human taeniasis carriers moving into the area?) I think the authors could expand on this point further.

PLOS authors have the option to publish the peer review history of their article (what does this mean?). If published, this will include your full peer review and any attached files.

Reviewer #1: Yes: Brecht Devleesschauwer

Reviewer #2: No
---

## [Decision Letter · Decision Letter 1]

3 Aug 2021

Dear Mr. Pray,

Thank you very much for submitting your manuscript "Validation of a spatial agent-based model for Taenia solium transmission (“CystiAgent”) against a large prospective trial of control strategies in northern Peru" for consideration at PLOS Neglected Tropical Diseases. As with all papers reviewed by the journal, your manuscript was reviewed by members of the editorial board and by several independent reviewers. In light of the reviews (below this email), we would like to invite the resubmission of a significantly-revised version that takes into account the reviewers' comments. 

There are a few issues arising which need to be addressed. Please see comments from the reviewer

We cannot make any decision about publication until we have seen the revised manuscript and your response to the reviewers' comments. Your revised manuscript is also likely to be sent to reviewers for further evaluation.

Sincerely,

Paul R. Torgerson

Associate Editor

Makedonka Mitreva

Deputy Editor

There are a few issues arising which need to be addressed. Please see comments from the reviewer

Reviewer's Responses to Questions

**Key Review Criteria Required for Acceptance?**

**Methods**

-Are the objectives of the study clearly articulated with a clear testable hypothesis stated?

-Is the study design appropriate to address the stated objectives?

-Is the population clearly described and appropriate for the hypothesis being tested?

-Is the sample size sufficient to ensure adequate power to address the hypothesis being tested?

-Were correct statistical analysis used to support conclusions?

-Are there concerns about ethical or regulatory requirements being met?

Reviewer #1: (No Response)

Reviewer #2: Thank you for you responses, they certainly helped and the improvements to the manuscript are noticeable. I have a few remaining questions to follow-up: 

In response to question & answer (1):

Original Q.1: It is not clear from the methods text, and tables 1 and Figure 1, how long pigs remain in the

“exposed” state, or how this is represented in the model. Does the represent the pre-patent period in pigs (before cyst maturation?)- from Table 1 and description in the methods section text, my impression is that pigs progress immediately after exposure based on the tuning parameter light-inf or heavy-inf without including a cyst maturation period in pigs? Please can the authors make this clearer in the text and describe the way progression from exposed to infected is parameterised in the model. 

New question:

Thanks for this. I think the section could do with a little more explanation/clarification, as an exposed compartment (as seen in Fig 1) would normally refer to individuals exposed but not infectious (in this context egg establishment and cyst maturation to a mature cyst, governed by a rate of transition). But I understand the exposed compartment, within this agent-based model formulation, represents an instantaneous transition governed by the light-all or heavy-all stochastic tuning (probability of exposure) parameters, and then an instantaneous transition to light infection (based on the probability of light infection upon contact parameter light-inf), or to heavy infection (based on the probability of heavy infection upon contact parameter heavy-inf). To this end, I think Figure 1 should be adjusted to provide more clarity on the transmission pathways: for example I think, if my understanding is correct, that there should be two pig exposure compartments, with light-all and heavy-all parameters to Exposed_light (EL) and Exposed_heavy (EH), which independently lead to the IL or IH compartments (and then two transitions from for EL and EH to IS based on light-sero and heavy-sero tuning parameters, which will improve clarity (if my understanding here is correct). Furthermore, I think Figure 1 could be improved by detailing the tuning parameters between compartment transitions, so that Figure 1 and Table 1 align more closely (also please see Figure 2 in the CystiSim Braae et et al. 2016 paper for example model schematic including transition probabilities within an IBM). 

In response to question & answer (3):

Original Q.3: I am a little unclear regarding the implications of lines 10-13 on page 11; so 42.9% are identified as infected if seropositive (30.1% with light infections plus 12.8% with heavy cyst infections), does this imply that 57.1% are false-positive or antibody positive from the EITB but not harbouring active larval infection (i.e. recovered)? 

New question: Thanks for this clarification, I think some text here with the relevant reference to briefly outline that the EITB is highly sensitive but not highly specific (hence the high false positive proportion due to measuring pigs that are exposed rather than only infected), would be useful in the first paragraph after Trial Outcomes (and to put the 56.1% FP proportion into context).

Minor:

Original Q:Can the authors provide a reference and explain the rationale for the cut-off for light infections

with <100 cysts and heavy infections with > 100 cysts (pg.6 lines 6-7). 

New question: Thanks, I think the authors may have missed a reference here in the updated text after “Exposure to eggs may cause light cyst infection (<100 cysts), and exposure to proglottid segments may lead to heavy cyst infection (≥ 100 cysts); either may lead to seropositivity” if this cut-off is based on data, or if not please state whether this is an assumption based on expert-opinion?

**Results**

-Does the analysis presented match the analysis plan?

-Are the results clearly and completely presented?

-Are the figures (Tables, Images) of sufficient quality for clarity?

Reviewer #1: (No Response)

Reviewer #2: Again, thank you for your responses which were very clear, and the corresponding edits to the manuscript. I have one remaining question:

Original Q. If I am understanding correctly, the median difference referenced on line 8 (for taeniasis) and 11 (porcine cysticercosis) of pg.14 relate to the median (absolute) differences? Can the authors explicitly state whether they refer to the absolute or relative differences throughout the manuscript please. 

New question: Thanks for the amendments and clarifications. I am not sure I fully understand the axis and accompanying results text for Figure 2. Does the x-axis on figure 2 for absolute error (bottom two panels) refer to (% points difference), as I am not sure how a median absolute error (difference) is therefore estimated from Figure 2 (i.e., the absolute error difference at baseline for taeniasis seems to have a max lower limit across villages of -0.03 and max upper limit across villages of +0.065)?

**Conclusions**

-Are the conclusions supported by the data presented?

-Are the limitations of analysis clearly described?

-Do the authors discuss how these data can be helpful to advance our understanding of the topic under study?

-Is public health relevance addressed?

Reviewer #1: (No Response)

Reviewer #2: (No Response)

**Editorial and Data Presentation Modifications?**

Reviewer #1: (No Response)

Reviewer #2: (No Response)

**Summary and General Comments**

Reviewer #1: Thank you for addressing my comments.

Reviewer #2: (No Response)

PLOS authors have the option to publish the peer review history of their article (what does this mean?). If published, this will include your full peer review and any attached files.

Reviewer #1: Yes: Brecht Devleesschauwer

Reviewer #2: Yes: Matthew Dixon-Zegeye
---

## [Editor Report · Decision Letter 2]

8 Oct 2021

Dear Mr. Pray,

We are pleased to inform you that your manuscript 'Validation of a spatial agent-based model for Taenia solium transmission (“CystiAgent”) against a large prospective trial of control strategies in northern Peru' has been provisionally accepted for publication in PLOS Neglected Tropical Diseases.

Best regards,

Paul R. Torgerson

Associate Editor

Makedonka Mitreva

Deputy Editor

I think the authors have satisfactorily addressed the reviewers comments and the manuscript can now go forward for publication

---

## [Editor Report · Acceptance letter]

22 Oct 2021

Dear Mr. Pray,

We are delighted to inform you that your manuscript, "Validation of a spatial agent-based model for *Taenia solium* transmission (“CystiAgent”) against a large prospective trial of control strategies in northern Peru," has been formally accepted for publication in PLOS Neglected Tropical Diseases.

Best regards,

Shaden Kamhawi

co-Editor-in-Chief

Paul Brindley

co-Editor-in-Chief
